# Molecular Characterization of Embryos with Different Buoyancy Levels in the Yellowtail Kingfish (*Seriola lalandi*)

**DOI:** 10.3390/ani12060720

**Published:** 2022-03-12

**Authors:** Phillip Dettleff, Javiera Rodríguez, Daniel Patiño-García, Renan Orellana, Rodrigo Castro, Sebastián Escobar-Aguirre, Ricardo Daniel Moreno, Jaime Palomino

**Affiliations:** 1Escuela de Medicina Veterinaria, Facultad de Agronomía e Ingeniería Forestal, Pontificia Universidad Católica de Chile, Santiago 8940000, Chile; pdettleff@udla.cl; 2Laboratorio de Reproducción Animal, Facultad de Ciencias Veterinarias y Pecuarias, Universidad de Chile, La Pintana, Santiago 8820000, Chile; javiera.rodriguez@veterinaria.uchile.cl; 3Departamento de Ciencias Fisiológicas, Facultad de Ciencias Biológicas, Pontificia Universidad Católica de Chile, Santiago 8320000, Chile; dpatino@uces.edu.co; 4Centro Integrativo de Biología y Química Aplicada (CIBQA), Universidad Bernardo O’Higgins, General Gana 1702, Santiago 8370854, Chile; renan.orellana@ubo.cl; 5Escuela de Medicina Veterinaria, Facultad de Recursos Naturales y Medicina Veterinaria, Universidad Santo Tomás, Talca 3460000, Chile; rodrigocastro@santotomas.cl; 6Facultad de Agronomía e Ingeniería Forestal, Pontificia Universidad Católica de Chile, San Joaquín, Santiago 8940000, Chile; sebastian.escobar@uc.cl; 7Escuela de Medicina Veterinaria, Facultad de Ciencias Médicas, Universidad Bernardo O’Higgins, Santiago 8320000, Chile

**Keywords:** Seriola, Yellowtail Kingfish, cathepsins, FAAs, pelagic fish, embryo

## Abstract

**Simple Summary:**

Low survival of embryos in *Seriola lalandi* captivity farming has been attributed to low buoyancy. This process is the result of oocyte hydration, which depends on the osmotic force exerted by free amino acids (FAA) generate of cathepsin-mediated yolk proteins proteolysis. In order to understand the molecular bases of buoyancy acquisition process and its loosing throughout the development of *S. lalandi*, the aim of this study was to compare gene expression and activity of cathepsins, as well as the FAA content between floating and low-floating embryos. Eggs, morula, blastula, gastrula and 24 h embryos were the stages collected in this study. These assessments were supplemented with morphometric and functional characterization of the embryos, where no differences in embryo and oil drop diameter, and constitutive gene expression were detected between floating and low-floating embryos. Cathepsin B did not show differences in expression or activity related to buoyancy condition. Both expression and activity of cathepsin D were higher in some low-floating developmental stages. By contrast, cathepsin L showed higher expression and activity in some floating early embryos. Higher FAA content was observed in floating embryos at least until gastrula stage in comparison to low-floating ones. In summary, expression and activity of cathepsins and FAA content, revealed specific pattern throughout development or buoyancy conditions of the embryos. This study identifies molecular differences between floating and low floating embryos at specific developmental stages where cathepsins and FAA are promising markers to evaluate the embryo quality in the farming of this species.

**Abstract:**

The buoyancy of eggs and embryos is associated with successful development in pelagic fish. Buoyancy is the result of oocyte hydration, which depends on the osmotic force exerted by free amino acids (FAA) generated by yolk proteolysis, and cathepsins are the main enzymes involved in this process. *Seriola lalandi* is a pelagic fish whose farming has been hampered by development failure that have been partially attributed to decreased buoyancy of embryos. Therefore, the aim of this study was to compare the mRNA expression and activity of cathepsins B, D, and L, as well as the FAA content in floating and low-floating embryos at different developmental stages. The chosen stages were eggs, morula, blastula, gastrula and 24 h embryos. Complementary assessments showed that there were no differences attributed to buoyancy status in embryo and oil droplet diameters, as well as the transcriptional status at any developmental stage. Cathepsin B did not show differences in mRNA expression or activity related to buoyancy at any stage. Cathepsin D displayed higher transcript and activity levels only in low-floating eggs compared with those floating. Cathepsin L showed higher expression in floating eggs and 24 h embryos compared with that of low-floating, but the activity of this enzyme was higher in floating eggs and morula. Total FAA content constantly decreased throughout development in floating embryos, but it was always higher than low-floating embryos until gastrula stage. In 24 h embryos floating and low-floating embryos share similar quantities of FAA. In summary, differences in the expression and activity of cathepsins between floating and low-floating embryos could be revealed at specific embryonic stages, suggesting different functions of these enzymes throughout development. Besides 24 h embryos, FAA content seems to be a decisive factor for buoyancy of embryos during early development of *S. lalandi*. Overall, considering the main role of cathepsins and FAA in buoyancy acquisition process and therefore in both embryo quality and viability, our study identifies good marker candidates to evaluate embryo quality in the farming of this species.

## 1. Introduction

In teleosts, early embryonic development occurs mainly inside the embryo coat and with limited nutrient exchange with the environment. Thus, embryo survival relies on yolk content, which is accumulated during egg maturation in the ovaries [1]. During follicular growth, yolk precursor proteins (e.g., vitellogenins) [2] are synthetized in the liver under 17 β-estradiol stimulation and then incorporated into the oocyte via receptor-mediated endocytosis [3]. Within the oocyte, the vitellogenins are enzymatically processed into lipovitellins, phosvitins, and β-components that are stored in yolk granules for future use during embryo development [4,5]. The cysteine proteases, cathepsins L and B, and the aspartic protease, cathepsin D, have been proposed as the main enzymes involved in vitellogenin processing in teleosts [6,7,8]. Additional processing of yolk proteins at the final stages of oocyte maturation has been documented in several marine species. This process generates free amino acids (FAA) and small peptides that are probably used as substrates for energy production via aerobic metabolism and protein synthesis during embryo development [2,9,10]. These FAAs also participate as osmotic effectors for oocyte hydration and subsequent cytoplasmic enlargement [11,12,13]. The extent of oocyte hydration varies among different teleost species, depending on their spawning behavior. While pelagic species spawn buoyant eggs with high hydration levels, benthic species produce slightly hydrated eggs, which remain mainly on the seafloor [13,14]. Therefore, the acquisition of buoyancy through hydration at the final stages of oocyte maturation represents a key step in the reproduction of pelagic fish, given that only floating eggs display normal embryo development [15]. Same as in the natural process, a positive buoyancy of embryo (i.e., embryos floating to the surface) is considered a quality indicator in hatchery reared pelagic fish, such as the Sea bream, *Sparus aurata* [16]; Japanese anguilla, *Anguilla japonica* [17]; Atlantic cod, *Gadus morhua* [18]; Common dentex, *Dentex* [19]; and Yellowtail kingfish, *Seriola lalandi* [20].

*Seriola lalandi* is a marine pelagic fish with worldwide distribution, and its importance for the aquaculture industry is growing [21,22]. Descriptions of gonadal morphology and steroidal plasma level determination in wild *S. lalandi* allowed the classification of this species as a multiple batch-spawner. Its reproductive period occurs during the austral spring and summer when the seawater temperature is over 17 °C [23]. Low survival of *S. lalandi* embryos has been frequently observed under captivity, hampering production scaling. This high mortality is partly due to the production of poor-quality eggs, which show a constant decrease in buoyancy during early embryonic development. Unfortunately, there is scarce information regarding molecular aspects involved in buoyancy acquisition and maintenance for eggs and embryos in *S. lalandi*. In a previous work, mRNA expression and activity of cathepsins in previtellogenic oocytes and floating developing embryos in *S. lalandi* [8]. Cathepsins were highly expressed in previtellogenic oocytes, but they decrease throughout development. Cathepsins activity displayed specific pattern where cathepsin B activity was stable until the blastula stage, unlike cathepsins D and L, whose high activity were observed in eggs than in later developmental stages [8]. These results help to understand the molecular aspects involved in egg yolk processing in *S. lalandi*, which subsequently affect the hydration process and embryo buoyancy level. Additionally, we recently reported that probably a caspase-mediated cell death pathway present in low-floating embryos might explicate the relation between the decrease of embryo survival with low buoyancy [24].

Considering that the acquisition of buoyancy relates upon cathepsin-mediate FAA generation, an unsolved question is that if the same process could be involved in buoyancy maintenance through early development. Thus, the aim of this study was to compare the mRNA expression and activity of cathepsins, as well as the FAA composition between floating and low-floating embryos during *S. lalandi* early development. Additionally, morphological and functional assessments of embryos were performed in order to know the possible effect of these aspects in the buoyancy condition of the embryos throughout the development.

## 2. Materials and Methods

All procedures were approved by the Ethics and Animal Care Committees of the main universities involved in this work. Protocols form Universidad Bernardo O’Higgins and Pontificia Universidad Católica de Chile (ID 190613051) were validated for the Research Ethics Committee of the Chilean National Foundation for Scientific and Technological Research.

### 2.1. Sample Collection

Samples for this study were collected from broodstock conditioned in three indoor tanks in the hatchery center of Acuinor SA Company, Caldera, Atacama Region, Chile. Each tank (2.5 m depth and 20,000 L capacity) contained 25 individuals in an expected male-to-female ratio of 2:1. Animals were subjected to thermal photoperiod management in which spawning events occurred spontaneously with temperatures above 19 °C and a photoperiod of 14 h. Feed consisted of 22 mm pellets Vitalis PRIMA Skretting for marine fish broodstock (54% protein, 18% lipid, 9.5% Ash, 0.9% fibre, 1.7% phosphorus, 7500 IU/kg Vitamin A, 1125 IU/kg Vitamin D3, 600 mg/kg Vitamin E and 1000 ppm Vitamin C). The diet was supplemented with fresh food (fisheries fish, squid, and cuttlefish) that were provided mainly during reproduction seasons in accordance with protocols of the company and the local fisheries availability. Embryos were collected during three spawning periods. Two independent batches from each tank (total six batches) were observed for spawning, and embryos were collected at different developmental stages as described previously [24]. Briefly, fertilized eggs were channeled from a skimmer on the surface of each tank into a separate egg collector tank, where embryo development was progressing (22–23 °C). The developmental stages collected were: fertilized eggs, morula, blastula, gastrula, and 24 h post fertilization embryos (24 h). After reaching the specified developmental stages, samples were collected from the egg collector tank and transferred to a conical inverted 4 L flask, and differences in the buoyancy levels of the samples were observed after 10 min. Exclusively floating samples were collected from the surface of the flask using a 500 μm mesh. Sinking embryos were discarded by opening a faucet located at the bottom of the flask and removing 200 mL of the water (with the sinking eggs). Using this faucet and a mesh, low-floating embryos were collected by extraction of 3 L of water. The samples were either stored in RNAlater Solution (Ambion^®^, Thermo Fisher Scientific, Waltham, MA, USA) for mRNA expression assessment using real-time polymerase chain reaction (qPCR) (50 individuals), frozen at −20 °C for the enzymatic activity assays and FAAs determination (100 individuals). Samples for morphometrics assessments were collected as explained below.

In order to standardize the enzymatic activity assays, ovary samples were obtained thorough cannulation of the gonophore of three anesthetized adult females during the spawning season (one female per tank). These tissues were frozen at −20 °C.

### 2.2. Experimental Design

Two experimental groups that correspond to floating and low-floating embryos were assessed at different developmental stages in each biological replicate, which corresponded to six independent spawning events (Figure 1).

### 2.3. Batches and Sample Characterization

The hatching rate (HR) of each spawning batch used in this study was quantified after 70 h of incubation of embryos in the egg collector tank (22–23 °C) using the morphological criteria described in this species [20,24,25]. The floating rate (FR) of each batch in the different developmental stages was registered in a 20 mL sub-sample obtained from the egg collecting tank, which was deposited in a 50 mL beaker where floating samples were isolated and their fraction recorded after 10 min. Additionally, to ensure that the data obtained in this study corresponded to the proposed developmental stage, a sub-sample of 20 eggs or embryos of each stage was fixed in 4% formaldehyde for evaluation under phase microscopy using a Leica DME phase contrast microscope (Leica Microsystems, Wetzlar, Germany). Only batches with morphological homogeneity [20,24,25], i.e., ≥70% of the individuals at the same stage of development (*n* = 14), were used for this study. Sample characterization also included the assessment of the diameters of individuals and their oil globules in floating and low-floating samples at each stage of the development. A millimeter ocular lens was used for these determinations at 100× magnification.

As *S. lalandi* embryos are highly auto-fluorescent, viability tools based on fluorescence activity would not be suitable in this study. Therefore, to indirectly estimate the viability of floating and low-floating embryos, the transcriptional status of these samples was estimated by mRNA expression assessment of four constitutive genes was compared among these samples (transcriptional status). These genes were beta-actin (*actb*), microtubule associated protein 1B (*map1b*), glyceraldehyde-3-phosphate dehydrogenase (*gapdh*), and 18S ribosomal RNA (*18S*). The procedures for these assessments are explained in Section 2.4.

### 2.4. RNA Isolation and Real-Time Polymerase Chain Reaction (qPCR)

Total RNA was extracted using the Gene JET RNA Purification Kit (Thermo Scientific, Eugene, Oregon, USA), according to the manufacturer’s instructions. The concentration of the total RNA was determined by fluorometric measurements in a Qubit^®^ Fluorometer using the Qubit^®^ RNA Assay Kit (Invitrogen^TM^, Eugene, OR, USA). DNA contamination was removed using DNase I treatment, and reverse transcription (RT) was performed using the enzyme conjugate SuperScript^TM^ First-Strand Synthesis System (Invitrogen, Carlsbad, CA, USA). Complementary DNA (cDNA) concentration was determined using the ssDNA Qubit^®^ Assay Kit (Molecular Probes^®^ Invitrogen, Eugene, OR, USA). Targeted (*catb*, *catd*, and *catl*) and constitutive genes (*actb*, *map1b*, *gapdh*, and *18S*) primers are showed in Table 1. These primers were designed and validated in a previous report according to standard qPCR procedures [8]. Using the Normfinder algorithm, both *actb* and *map1b* were selected as reference genes for expression normalization procedures according to their expression stability [8]. Prior to expression analysis of the target samples, we standardized and optimized the qPCR conditions for all primers using cDNA obtained from ovarian tissue to assess the optimal concentration for each primer pair and their PCR efficiency (E). We exclusively used primer pairs with an efficiency approaching 2 (100% efficiency) and R^2^ value > 0.989 [26].

Relative expression analysis of cathepsins B, D, and L in floating and low-floating embryos during the different developmental stages was performed using the 2^−ΔΔCt^ method [27]. Additionally, to compare the transcriptional status between floating and low-floating samples in all developmental stages included in this study, an assessment of threshold cycle value (Ct) of the four constitutive genes (*actb*, *map1b*, *gapdh*, and *18S*) was performed. Genes were amplified in an Eco Real-Time PCR System Model EC-100-1001 (Illumina^®^, San Diego, CA, USA) using the Maxima SYBR Green/Fluorescein qPCR Master Mix (Thermo Fisher Scientific, Waltham, MA, USA) following the manufacturer’s instructions. Control samples, without reverse transcriptase, cDNA template, or primers, were included in each plate.

### 2.5. Enzymatic Activity Assays

Crude extracts of ovarian tissue (4 mm^3^) and embryos (50 embryos per stage) were obtained by mechanical homogenization in 400 μL distilled water. The homogenates were centrifuged at 14,000× *g* for 10 min at 4 °C, and the supernatant was recovered for testing. The protein concentration was determined using the Qubit^®^ Fluorometer (Invitrogen^TM^_,_ Eugene, OR, USA) with the Qubit^®^ Protein Assay Kit (Molecular Probes^®^ Invitrogen). The activity of cathepsin B, D, and L was assessed using Abcam Fluorometric Activity Assay Kits (ab65300, ab65302, and ab65306 for cathepsins B, D, and L, respectively). Each assay was optimized in ovarian extracts. Control reactions that included background buffers and specific inhibitors were considered in each experiment. All assays were incubated at 37 °C for 60 min following the manufacturer’s instructions. Cathepsins activity was expressed as relative fluorescence units (RFU) per microgram of protein in the extracts.

### 2.6. Free Amino Acid (FAA) Determination

FAAs were determined throughout development in floating and low-floating embryos after conventional hydrolysis to release the constituent amino acids according to the method of Samaee et al. [19]. High-performance liquid chromatography (HPLC; HPLC Shimadzu, Japan), using an HPLC pump LC-20AD with diode-array detection (SPD-M20A detector; SIL-20A injector) was used to identify and quantify the amino acids. The derivatives were separated using an RP-18 (250 × 4.6 mm, 5–3 μm particle size; Inertsil^®^ ODS-3). Amino acid results are expressed as nanomoles of FAAs per milligram wet weight.

### 2.7. Statistical Analysis

Six biologicals replicates, representative of different batches from separated spawning events, were performed for all assessment of this work: In each replicate, nearly 5o in-dividual pool at each developmental stage and buoyancy state were used for mRNA extraction for cathepsins and constitutive genes expression assessments. Another 50 individual pool was used for cathepsin activity assays and a similar pool for FAA determination. For morphometric evaluations (embryo and oil drop diameter) pools of 20 individuals were used at each biological replicate. Data were analyzed using ANOVA (InfoStat Professional Program, National University of Córdoba, Argentina). Each model included the main effects of the developmental stages, the buoyancy condition, and their interaction. Significant differences between means were evaluated using Tukey’s test. All values were considered significantly different for *p* < 0.05 [28].

## 3. Results

### 3.1. Characterization of Batches and Samples

The six batches assessed in this work displayed an average HR of 85.7 ± 17.5% (mean ± standard deviation (SD) after 70 h of incubation. The FR estimated at each developmental stage did not show significant differences until the blastula stage with an FR value > 80% (Figure 2A). The FR at the gastrula and 24 h stages was lower than that obtained in eggs and morula; however, it did not differ from that observed in blastula (Figure 2A). The newly fertilized eggs and the different embryo stages assessed in this study were transparent and spherical, with average diameters of 1.40 ± 0.04 mm and 1.38 ± 0.03 mm in floating and low-floating samples, respectively. However, no significant differences were found in the diameters of embryos in different developmental stages or buoyancy levels (Figure 2B). All eggs and early embryos possessed a single oil drop with an average diameter of 0.35 ± 0.03 and 0.32 ± 0.03 mm in floating and low-floating samples, respectively. As well as for total embryo diameter, no significant differences were observed in oil droplet diameter among the samples (Figure 2C).

### 3.2. Transcriptional Status of the Samples

A similar transcriptional status between floating and low-floating embryos could be assumed as no significant differences in Ct values were detected for any constitutive genes in all developmental stages analyzed in this study (Figure 3). *18S* gene showed high mRNA levels throughout *S. lalandi* development, with average Ct values below 20 (Figure 3D). Transcription amounts with Ct values between 20 and 30 were observed for *actb*, *gapdh*, and *map1b* (Figure 3A–C, respectively). The *actb* and *map1b* mRNAs were expressed without variation from egg to the gastrula stage; however, in the 24 h embryos, a lower level of these transcripts was detected compared with that of the other developmental stages. The *gapdh* mRNA expression decreased as development progressed, and *18S* did not show a clear expression profile trend.

### 3.3. Cathepsin mRNA Expression Analysis

RT-qPCR analysis was used to determine mRNA levels encoding *catb*, *catd*, and *catl* in floating and low-floating eggs and embryos of *S. lalandi* (Figure 4). *Catb* mRNA abundance did not show differences attributable to buoyancy throughout development (Figure 4A). However, the levels of this transcript were higher in eggs, morula, and blastula than in more advanced stages (Figure 4A). *Catd* mRNA levels were higher in low-floating than in floating samples only at the egg stage (Figure 4B, asterisk). Comparatively, floating blastula embryos showed a higher level of *catd* mRNA only when compared with 24 h embryos. Low-floating eggs and morula showed similar levels to the blastula or gastrula, but higher levels of *catd* than 24 h embryos (Figure 4B). *Catl* mRNA levels were similar at all studied stages in low-floating samples (Figure 4C). *Catl* mRNA was more highly expressed in floating eggs than in all other stages regardless of the buoyancy state. In floating samples, this transcript did not show differences from the morula up to the 24 h stage (Figure 4C). However, *catl* mRNA was higher in floating than in low-floating samples.

### 3.4. Enzymatic Activity

Ovarian-extract enzymes showed significantly higher activity than that of the blank, and was dramatically reduced when an inhibitor of cathepsin was present in the assays (Figure 5). These results suggest that the values obtained were specific for the respective enzymes. Cathepsin B activity showed a similar pattern throughout development in both floating and low-floating and embryos, with a gradual decrease with the increase in embryogenic complexity (Figure 5A). Differences in cathepsin B activity related to sample buoyancy were not detected at any stage (Figure 5A). Cathepsin D activity was higher in low-floating than in floating samples only at the egg stage, and then it sharply decreased when embryos reached morulae and then remained constant from blastula to 24 h embryos (Figure 5B). Regarding the floating samples, we determined a significant reduction in cathepsin D activity when embryos reached the 24 h stage (Figure 5B). Cathepsin D activity was higher in low-floating than in floating samples only at the egg stage, and then sharply decreased when embryos reached morula stage and remained constant from blastula to 24 h embryos. In the other developmental stages, no differences were observed with respect to buoyancy (Figure 5B). The highest levels of cathepsin L activity in floating samples were observed in the egg, morula, and blastula stages, without differences among them. In low-floating embryos, cathepsin L activity showed a progressive decrease after the blastula stage; the lowest activity level was detected in 24 h embryos compared with the first three developmental stages (Figure 5C). Differences in cathepsin L activity between floating and low-floating embryos were only detected in the egg and morula stages; the floating samples presented higher enzymatic activity than low-floating samples (Figure 5C).

### 3.5. FAA Composition

Total FAA content in floating and low-floating embryos at different stages of *S. lalandi* development is presented in Table 2. In floating samples, total FAAs gradually decreased throughout development; however, it was significantly lower in 24 h embryos compared with eggs (Table 2). In low-floating embryos, there were no differences in total FAA content through development. Differences between floating and low-floating samples in total FAA content were observed until the gastrula stage; the total FAA content was higher in floating than in low-floating samples (Table 2). Non-essential FAAs (NFAAs) displayed a similar content pattern to those observed in total FAAs. Essential FAAs (EFAAs) showed the same content pattern observed in NFAAs; however, EFAA content was higher in floating embryos than in low-floating embryos. Overall, the most abundant NFAAs were Serine and Alanine, which were also the most abundant in floating embryos until the gastrula stage (Table 2). Leucine and Lysine were the most abundant EFAAs in all samples (Table 2).

## 4. Discussion

In this study, we observed a decrease in the floatability percentage throughout development. A decrease in buoyancy through early development has been documented in other marine pelagic fish, such as the Atlantic cod, *G. morhua* [18,29]; European anchovy *Engraulis ecrancicolus* [30]; and Blue whiting *Micromesistius poutassou* [31]. Buoyancy loss has been associated with a slight increase in specific embryo gravity. Therefore, the presence of embryos with reduced buoyancy or increased density is a common situation in pelagic species; this has been mainly attributable to passive water loss from the embryo to the marine, hyperosmotic environment. Water loss occurs until the osmoregulation mechanism is fully established, which happens after gastrulation in fish [32,33]. This process has not been investigated in *S. lalandi* embryo development; however, considering our results, in which buoyancy decrease was observed until 24 h embryos, we can infer that osmoregulation mechanisms are activated in more advanced developmental stages.

In this study, low-floating embryos displayed morphological parameters similar to those observed in floating embryos, and no statistical differences in the diameters of whole embryo or oil droplets were observed. In addition, a similar transcriptional status between floating and low-floating samples could be found, as no differences in constitutive-gene expression were detected between these samples. Therefore, the differences attributable to buoyancy in *S. lalandi* embryos could be associated with the mechanism involved in the hydration process, which starts in the ovary but ends in the first stages of embryo development [2,6,34,35].

In this study, we confirmed previous results, showing higher levels of mRNA and activity of cathepsin B in very early stages in *S. lalandi* (e.g., eggs, morulae, and blastula) than in later stages (e.g., gastrula and 24H) [8]. A similar mRNA expression pattern and activity of this enzyme was reported in the pelagic Red Spotted Grouper, *Ephinephelus akaara* [36] and *D. labrax*, respectively [6]. Opposing results have been described in Mummichog *Fundulus heteroclitus* embryogenesis, in which both transcripts and activity of cathepsin B gradually increased in embryos from the 2- to 4-cell stage up to 7 days post fertilization [35]. This difference could be attributed to the different spawning strategies of *F. heteroclitus*, which produces benthic eggs. In this species, it has been suggested that cathepsin B may be involved in the mechanisms underlying the onset of gastrulation, playing complementary roles with other cathepsins [35]. Therefore, considering that in this study no differences in both mRNA and activity of cathepsin B were detected between floating and low-floating embryos at any developmental stage, it could be hypothesized that this enzyme is not directly involved in yolk proteolysis, hydration process, or the subsequent buoyancy in *S. lalandi*. Indeed, both mRNA expression and activity patterns in pelagic species corroborate the proposed role for cathepsin B in this species, which would act as an activator for the aspartic protease cathepsin D [6]. In this context, its presence and activity could be more important in the early stages of embryo development.

Cathepsin D is involved in the initial hydrolysis of vitellogenin to generate phosphovitin, lipovitellin, and β-components, which act as the first osmotic vectors required for egg hydration [3,6]. This process occurs within yolk granules in a controlled manner, in which the enzymatic activity is regulated by different mechanisms, such as inhibitors, activators, or enzymes at the zymogen state [6]. This controlled process would allow a progressive degradation of yolk content during embryonic development, generating a constant production of osmotic vectors for water influx and, subsequently, permanent FAA supply. Interestingly, cathepsin D showed higher mRNA levels and activity in low-floating than in floating eggs, which was not observed in more advanced developmental stages. This result is consistent with the high activity of this enzyme reported in non-floating eggs of *D. labrax* [6]. Nevertheless, cathepsin D has been identified as a mediator enzyme in the programmed cell death process during ovarian remodeling in teleost fish [37], and its involvement in apoptotic mechanisms has been proposed in *S. aurata* non-floating eggs [38] and *S. lalandi* low-floating early embryos [24]. These findings support the hypothesis that pelagic-species eggs and embryos that lose their buoyancy may have premature death, and that precipitant expression and activation of cathepsin D would accelerate this process. Therefore, cathepsin D may be considered as a possible marker for poor-quality eggs in *S. lalandi*, as has been suggested in *S. aurata* and *D. labrax* [6,15].

In contrast, *catl* mRNA was more abundant in floating than in low-floating eggs. This difference was also detected in 24 h embryos. In floating eggs and morula, cathepsin-L activity was higher than that in more advanced stages and in all low-floating samples, respectively. This enzymatic activity pattern is congruent with that observed for embryo buoyancy in *D. labrax* [6]. It has been proposed that cathepsin L participates in the proteolysis of vitellogenin residues, specifically high molecular weight lipovitellin. The result is FAAs that act as osmotic effectors for buoyancy maintenance and also as substrates for energy generation during embryo development [10]. Therefore, high levels of transcripts and enzymatic activity of cathepsin L detected in floating early developmental stages (eggs and morula) are compatible with cathepsin L involvement in the second protolithic cleavage of lipovitellin components [10]. Therefore, cathepsin L could be considered a possible good quality marker, but only in the early stages in *S. lalandi*.

Free amino acids resulting from the proteolytic cleavage of yolk proteins have been proposed as a resource for energy production, protein synthesis, and osmotic effectors for hydration in pelagic fish embryos [39,40,41]. In this study, the most dominant essential FAAs during embryo development were leucine and lysine, while alanine and serine dominated among the non-essential FAAs. These results corroborate findings reported in *S. lalandi* [40]. Moreover, similar results are described in other pelagic species, in which usually more than one FAA is predominant with little variation in their importance [19,42,43]. However, benthic species embryogenic development has a profile dominated by a single FAA [44]. Another interesting result from our study is that in the FAA profile during development in floating embryos, the total FAA content decreases as development progresses, as verified for other pelagic fish [40,41,43,45]. In pelagic species, the importance of metabolic fuel for energy has been attributed to FAAs during embryo development [9,40]. Thereby, the decreasing in the FAAs content with the development progressing observed in floating embryos, is consistent with the loss of buoyancy of embryos as it was demonstrated in this work and confirms previous results in *S. lalandi* where the loss of positive buoyancy around the time of hatch was also associated with the decreasing in FAAs content during embryogenesis [40].

In summary, this study shows that molecular differences between floating and low-floating embryos could be revealed at specific early embryonic stages. Our results suggest that cathepsin biological functions probably change according to the developmental stage. Therefore, considering the transcripts and activity patterns of cathepsin D and L, these enzymes could be good candidates for markers to evaluate embryo quality in aquaculture. Finally, with the exception of 24 h embryos, differences related to buoyancy were detected in FAA content during development in *S. lalandi*; FAA profile throughout development observed in floating samples fits the profile proposed for pelagic species. Therefore, FAA content could also be considered as a quality marker during early development of *S. lalandi*.

## Figures and Tables

**Figure 1 animals-12-00720-f001:**
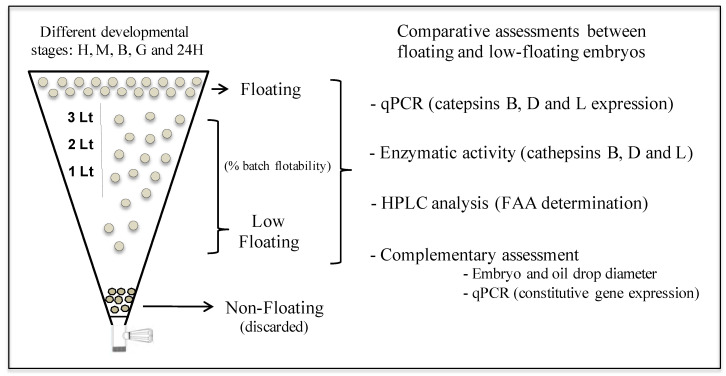
Experimental design. The developmental stages H, M, B, G and 24H correspond to fertilized eggs, morula, blastula, gastrula, and 24 h post fertilization embryos, respectively.

**Figure 2 animals-12-00720-f002:**
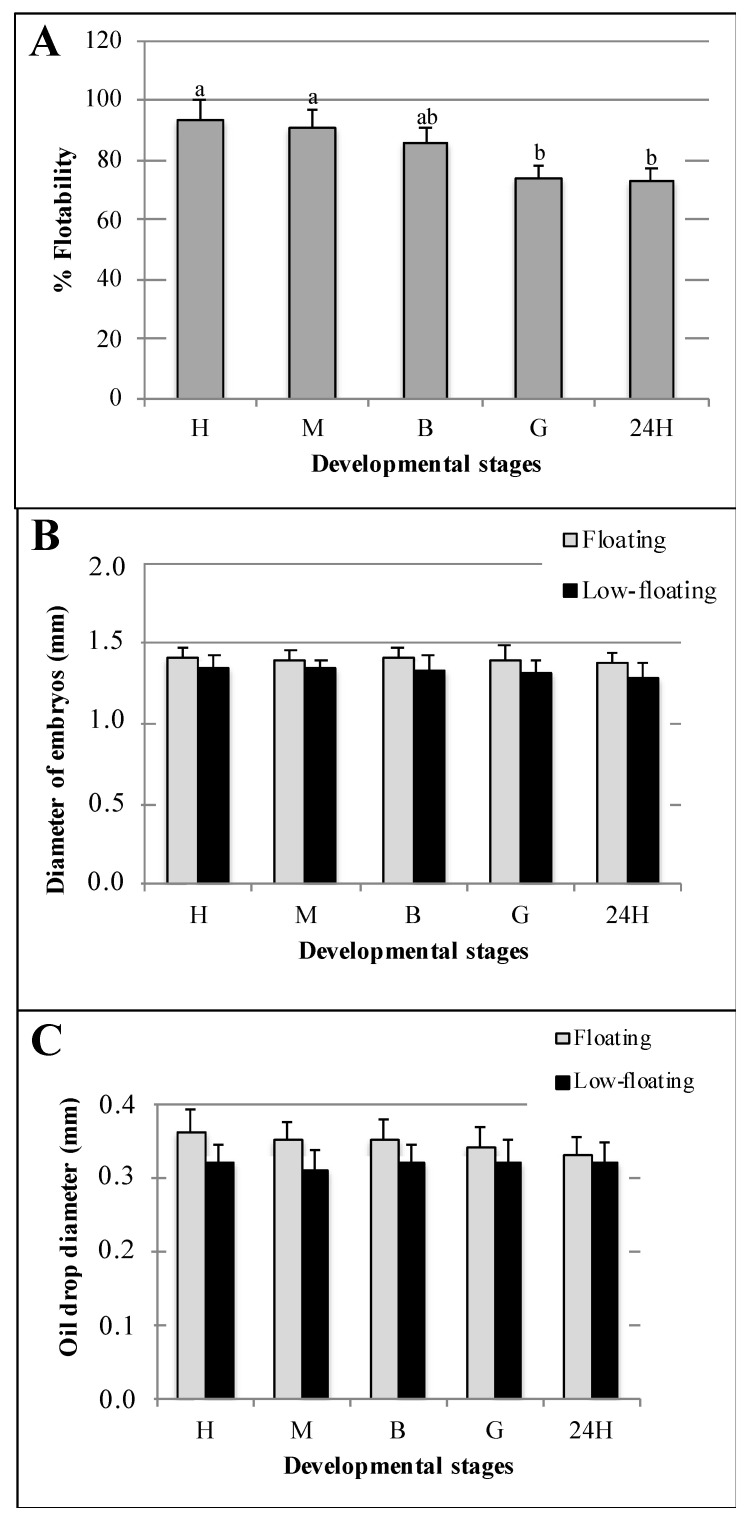
Morpho-biometric parameters of batches (**A**) and embryos (**B**,**C**) assessed in this study. Buoyancy level (% Floatability) observed in each developmental stage (**A**). Total embryo (**B**) and oil drop (**C**) diameters observed in both floating (gray bars) and non-floating (black bars) embryos. Fertilized eggs, morula, blastula and 24 h post-fertilization embryos are identified as E, M, B, G and 24H, respectively. Bars represent mean ± standard deviation (SD) of six biologicals replicates, and the different letters indicate significant differences (*p* < 0.05) assessed using Tukey’s test.

**Figure 3 animals-12-00720-f003:**
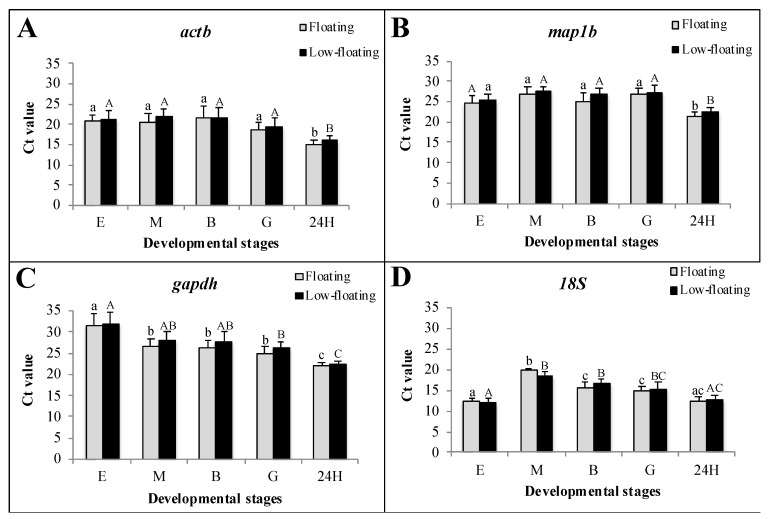
Transcriptional status of floating and low-floating embryos throughout development, assessed using the Ct value of four constitutive genes. Expression of *actb*, *map1b*, *gapdh* and *18S* are showed in the graphs (**A**), (**B**), (**C**) and (**D**), respectively. Fertilized eggs, morula, blastula and 24 h post-fertilization embryos are identified as E, M, B, G and 24H, respectively. Bars represent mean ± standard deviation (SD) of six biologicals replicates, and different lowercase or uppercase letters indicate statistically significant differences in floating or low-floating embryos, respectively (*p* < 0.05).

**Figure 4 animals-12-00720-f004:**
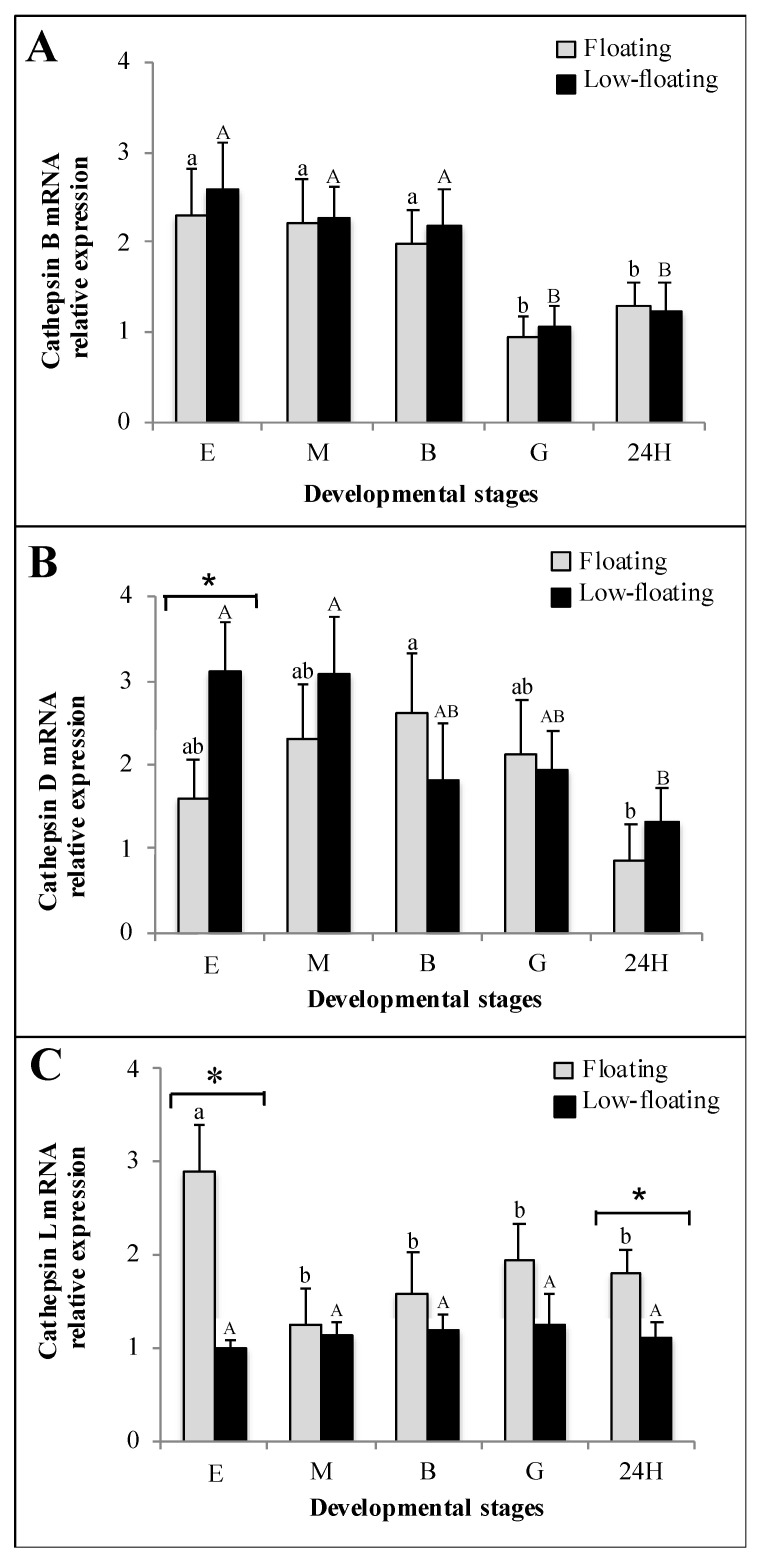
Relative expression of cathepsins B (**A**), D (**B**) and L (**C**), throughout *Seriola lalandi* development in floating (gray bars) and low-floating (black bars) embryos, assessed through RT-qPCR. Fertilized eggs, morula, blastula and 24 h post-fertilization embryos are identified as E, M, B, G and 24H, respectively. Bars represent mean ± standard deviation (SD) of six biologicals replicates, and different lowercase or uppercase letters indicate statistic differences within floating or low-floating embryos, respectively (*p* < 0.05). Differences between floating and low-floating embryos are identified with an asterisk over a bracket.

**Figure 5 animals-12-00720-f005:**
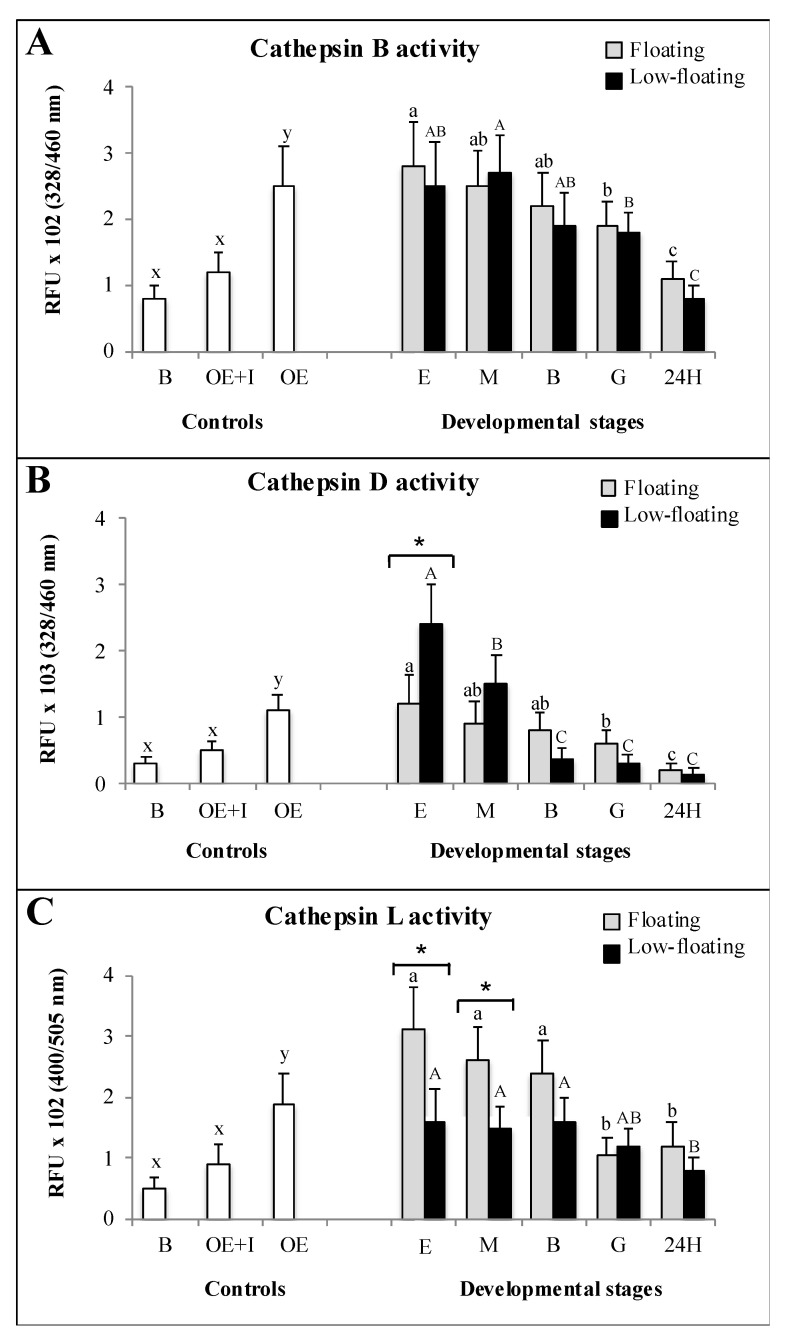
Enzymatic activity assays of cathepsins B (**A**), D (**B**) and L (**C**) throughout *Seriola lalandi* development in floating (gray bars) and low-floating (black bars) embryos. White bars represent control reaction samples in which negative controls with the reaction buffer were considered (RB). The specificity of reactions was determined by adding specific inhibitors to ovarian extracts (OE + I). Fertilized eggs, morula, blastula and 24 h post-fertilization embryos are identified as E, M, B, G and 24H, respectively. Bars represent mean ± standard deviation (SD) of six biologicals replicates, and different lowercase or uppercase letters indicate statistically significant differences within floating or and low-floating embryos, respectively (*p* < 0.05). Differences between floating and low-floating embryos are identified with an asterisk over a bracket.

**Table 1 animals-12-00720-t001:** Sequences of qPCR primers.

Gene	Primer Sequences (5′-3′)	TM (°C)	Efficiency	Amplicon (bp)
*catb*	F: GTAATGGTGGCTACCCTTCA	54.3	1.97	279
	R: CATACTGAATCTGCTCCTCG	52.5		
*catd*	F: GCCAAGTCCAGCACATACG	56.3	2.19	215
	R: ACAGAGATGCGTGGGTAGG	56.6		
*catl*	F: ACTACAACTCTGCCAACGAC	54.8	1.97	165
	R: AACTGGAAAGACTCGTGACC	54.6		
*actb*	F: AGGGAAATCGTGCGTGACAT	57	2.04	563
	R: GCTGAAGTTGTTGGGCGTTT	56.7		
*map1b*	F: TCATCAAGATTATCAGGAGGCG	54.3	1.98	158
	R: GGAAGCATACACCATGTAGAGG	55		
*gapdh*	F: CCCTTCATCGACCTGGAGTA	55.8	1.99	459
	R: GAGCAGAGGCCTTCTCAATG	55.6		
*18S*	F: GCTCGTAGTTGGATCTCGGG	57.3	2.01	597
	R: GGTGAGGTTTCCCGTGTTGA	57.5		

**Table 2 animals-12-00720-t002:** Qualitative and quantitative profile of FAAs in floating and low-floating *Seriola lalandi* developing embryos. Fertilized eggs, morula, blastula and 24 h post-fertilization embryos are identified as E, M, B, G and 24H, respectively. Value represents the FAA amount (nanomoles of FAAs/mg wet weight) ± standard deviation (SD) of six replicates, and the different letters in rows indicate significant differences (*p* < 0.05) assessed using Tukey’s test. NEFAAs correspond to non-essential FAAs and EFAAs to essential ones.

		Floating Developing Embryos					Low-Floating Developing Embryos			
	H	M	B	G	24H	H	M	B	G	24H
**Nonessential FAAs**										
Aspartic acid	0.71 ± 0.06	0.55 ± 0.04	0.53 ± 0.02	0.49 ± 0.03	0.41 ± 0.02	0.43 ± 0.02	0.26 ± 0.03	0.50 ± 0.03	0.44 ± 0.03	0.39 ± 0.02
Glutamic acid	1.91 ± 0.11	1.77 ± 0.09	1.56 ± 0.15	1.45 ± 0.13	1.12 ± 0.08	0.96 ± 0.09	1.06 ± 0.02	1.06 ± 0.08	0.95 ± 0.10	0.81 ± 0.09
Serine	**7.37 ± 1.02 ^a^**	**7.38 ± 0.96 ^a^**	**6.64 ± 0.82 ^ab^**	**6.42 ± 0.87 ^ab^**	**5.21 ± 0.49 ^bc^**	**4.91 ± 0.40 ^c^**	**4.12 ± 0.34 ^d^**	**4.61 ± 0.55 ^cd^**	**4.75 ± 0.58 ^cd^**	**4.80 ± 0.64 ^cd^**
Gycine	2.43 ± 0.42	2.38 ± 0.39	2.31 ± 0.45	2.16 ± 0.34	1.54 ± 0.21	1.62 ± 0.15	1.29 ± 0.07	1.52 ± 0.11	1.62 ± 0.16	1.40 ± 0.09
Alanine	**9.57 ± 2.35 ^a^**	**8.44 ± 2.54 ^a^**	**7.68 ± 1.87 ^a^**	**7.73 ± 2.21 ^a^**	**7.03 ± 1.95 ^ab^**	**5.68 ± 0.75 ^b^**	**4.72 ± 0.63 ^b^**	**5.21 ± 0.75 ^b^**	**5.95 ± 0.85 ^b^**	**5.35 ± 0.85 ^b^**
Proline	1.82 ± 0.14	1.77 ± 0.12	1.67 ± 0.14	1.63 ± 0.08	1.41 ± 0.09	1.26 ± 0.10	1.09 ± 0.06	1.14 ± 0.09	1.22 ± 0.08	1.06 ± 0.07
Cysteine	0.60 ± 0.03	0.41 ± 0.03	0.28 ± 0.03	0.16 ± 0.02	0.14 ± 0.02	0.07 ± 0.02	0.29 ± 0.09	0.15 ± 0.03	0.12 ± 0.02	0.11 ± 0.02
**∑ NEFAAs**	**24.41 ± 4.13 ^a^**	**22.00 ± 4.20 ^ab^**	**20.67 ± 3.21 ^ab^**	**20.04 ± 3.68 ^ab^**	**16.86 ± 2.86 ^bc^**	**14.92 ± 1.53 ^c^**	**12.80 ± 1.24 ^c^**	**14.20 ± 1.64 ^c^**	**15.04 ± 1.82 ^c^**	**13.93 ± 1.78 ^c^**
**Essential FAAs**										
Histidine	1.43 ± 0.07	1.38 ± 0.09	1.33 ± 0.07	1.26 ± 0.05	0.86 ± 0.04	0.89 ± 0.02	0.77 ± 0.02	0.89 ± 0.03	0.94 ± 0.05	0.81 ± 0.02
Arginine	3.36 ± 0.38	3.35 ± 0.41	3.10 ± 0.35	2.98 ± 0.28	2.22 ± 0.20	2.23 ± 0.22	1.81 ± 0.12	2.06 ± 0.18	2.21 ± 0.20	2.14 ± 0.21
Threonine	3.11 ± 0.25	3.17 ± 0.30	2.79 ± 0.25	2.72 ± 0.28	2.24 ± 0.22	2.10 ± 0.17	1.92 ± 0.10	2.02 ± 0.20	2.13 ± 0.17	1.94 ± 0.09
Tyrosine	1.82 ± 0.18	1.50 ± 0.17	1.43 ± 0.15	1.39 ± 0.08	1.14 ± 0.09	1.11 ± 0.06	0.99 ± 0.06	0.97 ± 0.08	1.07 ± 0.05	0.95 ± 0.06
Valine	5.15 ± 0.65	5.09 ± 0.60	4.62 ± 0.54	4.44 ± 0.41	4.23 ± 0.33	3.37 ± 0.35	3.22 ± 0.30	3.51 ± 0.39	3.41 ± 0.39	3.26 ± 0.38
Methionine	1.70 ± 0.12	1.70 ± 0.15	1.45 ± 0.12	1.34 ± 0.09	0.99 ± 0.05	1.00 ± 0.05	1.04 ± 0.04	0.98 ± 0.07	0.94 ± 0.05	0.72 ± 0.04
Isoleucine	5.18 ± 0.77	4.97 ± 0.85	4.44 ± 0.83	4.30 ± 0.63	4.17 ± 0.45	3.46 ± 0.41	3.21 ± 0.35	4.14 ± 0.50	3.55 ± 0.45	3.13 ± 0.38
Leucine	**7.82 ± 1.55 ^a^**	**7.77 ± 1.98 ^a^**	**7.44 ± 2.04 ^a^**	**6.69 ± 1.60 ^ab^**	**6.07 ± 1.34 ^ab^**	**5.08 ± 1.24 ^b^**	**4.88 ± 0.58 ^b^**	**5.07 ± 1.54 ^b^**	**5.24 ± 1.47 ^b^**	**4.78 ± 0.55 ^b^**
Phenylalanine	2.57 ± 0.16	2.13± 0.15	2.09 ± 0.16	2.03 ± 0.12	1.76 ± 0.12	1.55 ± 0.06	1.31 ± 0.07	1.96 ± 0.14	1.50 ± 0.06	1.43 ± 0.06
Lysine	**7.02 ± 1.56 ^a^**	**6.35 ± 1.36 ^a^**	**6.16 ± 1.56 ^a^**	**5.74 ± 1.46 ^ab^**	**5.27 ± 1.20 ^ab^**	**4.46 ± 0.94 ^bc^**	**3.68 ± 0.45 ^c^**	**4.60 ± 1.35 ^bc^**	**4.16 ± 0.47 ^bc^**	**4.14 ± 0.48 ^bc^**
**Total EFAAs**	**39.16 ± 5.69 ^a^**	**37.40 ± 6.06 ^ab^**	**34.85 ± 6.07 ^ab^**	**32.89 ± 5.01 ^ab^**	**28.95 ± 4.14 ^bc^**	**25.25 ± 3.48 ^cd^**	**22.82 ± 2.09 ^cd^**	**26.21 ± 5.02 ^cd^**	**25.15 ± 3.36 ^cd^**	**23.29 ± 2.27 ^d^**
**Total FAAs**	**63.57 ± 9.82 ^a^**	**59.40 ± 10.26 ^ab^**	**55.52 ± 9.28 ^ab^**	**52.93 ± 8.69 ^ab^**	**45.85 ± 7.00 ^bc^**	**40.17 ± 5.01 ^c^**	**35.62 ± 3.33 ^c^**	**40.41 ± 6.66 ^c^**	**40.19 ± 5.18 ^c^**	**37.22 ± 4.05 ^c^**

## Data Availability

“Data available on request from the authors”. The data that support the findings of this study are available from the corresponding author upon reasonable request.

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
