# Peer review of "Molecular Characterization of Embryos with Different Buoyancy Levels in the Yellowtail Kingfish (Seriola lalandi)"

_animals, 2022, doi:10.3390/ani12060720_

Round 1
Reviewer 1 Report
Dettleff et al., provide many morphological and molecular based comparisons in high-floating vs low-floating embryos. The paper has some flaws in terms of how the objectives and methods are presented as well as how the data were analyzed. Overall this paper is also redundant with work some of the authors have published on the same species. They clearly have put a bunch of time and energy into this project, I think that if the authors spend some time clarifying their objectives, methodology, and data
analyses they have a paper that is worth publishing. My overall recommendation accept after major revision.
I wonder if this manuscript is a bit redundant with two of the articles they cite: Palomino, J.; Herrera, G.;Dettleff, P.; Patel, A.; Torres-Fuentes, J.L.; Martínez, V. Assessment of cathepsin mRNA expression and enzymatic activity during early embryonic development in the yellowtail kingfish Seriola lalandi. Anim.Reprod. Sci. 2017, 80, 23-29. doi: 10.1016/j.anireprosci.2017.02.009.
Palomino, J.; Gómez, C.; Otarola, M.; Dettleff, P.; Patiño-García, D.; Orellana, R.; Moreno, R.D. Embryobuoyancy and cell death markers in yellowtail kingfish (Seriola lalandi Valenciennes 1833) during earlyembryogenesis. Front. Cell. Dev. Biol. 2021, 9, 630947. doi: 10.3389/fcell.2021.630947. I understand that this paper is meant to build off those studies but the second is not even referenced in the introduction as pertinent background information.
The authors really need to clearly state their objectives in the introduction as they seem to have many objectives (some of which overlap with previous papers), but none of the objectives are clearly defined.
I think it would help them layout the blueprint for the results and discussions if they just clearly stated objectives a, b, and c rather than trying to cram everything into single, unclear sentences. I also think they should create a flow chart for their methodology so that their methodology to address each hypothesis is clear. As of now the methodology is about as confusing as their objectives.
Their data analyses are also not much better, as they are brief in this section and do not address how they are dealing with their design which very much seems like a repeated measures design and with very few samples. I have reservations of their “significance” based on how small the sample size is. They did not state a single alpha value for any comparison and there is no mention if they used an adjusted p-value for multiple pairwise comparisons. I also wonder if a time series type analyses might be more appropriate (especially when concerning the FAA comparisons) as these embryos cannot make more FAAs at this stage.
Easily fixed inconsistencies:
The authors jump between tenses (past and present) during the paper (introduction, methods).
The authors treat common names as nouns (no capitalization of each word) and proper nouns (capitalization of each word). The AFS has stated a preference for proper nouns but I think that suggestion is poor. They also state a common name for some species and not others. Whatever the
authors decide, be consistent.
Use of hyphens between different cathepsins.
Specific:
L4: Seems odd the first author is numeral 2 and not 1.
Simple Summary
L19: Replace attributable to attributed
L23: Delete in after “aimed”.
L22-24: This sentence is unclear as written likely because of the three commas and this sentence going
on with poor verb associations. As written I am unclear if gene expression and activity is also being associated with FAAs or if FAAs are being used to understand the molecular bases of buoyancy. Please just revise this entire sentence so that the objective is clear and concise.
L28: replace until gastrula stage with “(at least until the gastrula stage)”.
L32: I think I understand this sentence, but I am uncertain. Are they saying that these markers should be used to determine the quality of embryos produced in aquaculture or that they saying they used them to identify egg quality… I think the “in which” on L31 is what is confusing the sentence.
Abstract
L36-37: Perhaps replace “, which is partly due to decreased buoyancy” to “; failures that have been partially attributed to decrease buoyancy”.
L39-40: Here again exactly what they did (objectives) are unclear. I suggest thte authors not try to save 5 words, but instead specifically state the objectives. They are measuring something related to mRNA (no
clue what in current context), activity profiles of three cathepsins, and overall FAA content (not sure)?
L40-41: I think this sentence is backwards. No difference in buoyancy was attributed to biometric parameters or transcriprional activity. Also, what biometric parameters?
L42 differences in mRNA what? Sequences? Morphology, amount?
L44: What other stages? I am lost.
L44-45: At no prior point do the authors state that they are looking into different embryonic stages…
floating eggs vs 24 h embryos.
L46 replace “and” with “but”?
L52-53. More context is needed here as above.
Introduction
L63: delete the comma after components and replace which with that.
L68: delete the comma after peptides and replace which with that.
L77-79: I suggest “ Additionally, positive buoyancy for embryos (i.e., embryos floating to the surface) is
considered a quality indicator in hatchery reared pelagic fishes”
L81: Why the authority for S. lalandi and not others?
L83: Replace for with “within”.
L86: No comma needed after summer
L91: Replace in with “for”.
L92: This is an odd place to cite paper 8…. At first I thought the “We” referred to the authorsof this paper rather
than a citation of previous work.
L93-100: They should really summarize what these results were rather than make it seem like the discussion or abstract from that particular paper. Seem like that section could be reduced into a single sentence that would flow
better with the statement of objectives for this study.
L104-107: “Would allow the proposition” is poorly phrased. This entire sentence seems out of plae as it seems like this is justification for their objectives and should come before.
L108-109: This sentence/objective sort of came out of nowhere as it wasn’t in the short summary or abstract and I am honestly a bit confused on what they mean throughout this sentence… “morpho-biometric antecedents of embryos” is not clear.
Materials and Methods
L111-113: Perhaps this is journal style but I feel this could be in acknowledgements. Are numbers
associated with these protocols as they are in other countries?
L118: How do you get a 2:1 ratio and have 25 fish? 16 males: 8 female and 1 unknown? Do all 8 females
spawn at the same time?
L124: Using “us” here is odd as the authors are not all the same.
L124-128: So were stages collected or did they collect fertilized eggs and then allow those eggs to
develop?
Section 2.2: The authors should really include a flow chart for collection, treatment, and end point analyses. This would help clarify exactly what their methodology was to address each of their objectives.
As is written section 2.2 is quite confusing because I am not following if “this study” means the overall study or a subsection of the study given section 2.1 deals with a portion of “this study”. I also think the second paragraph of this section should just be in 2.3.
L170: Why did they choose these genes?
L172: Reference should be a number.
L177-178: Not certain if the “we” here refers to the reference where they had this success in the past
(citation) or if it was in this study.
Table 1. Could be in supplemental.
2.4: The number of replicates should be included here (or in a flow chart figure).
L217: The authors cite Samaee et al. 2010 as Ref [19] so “(2010)” can be deleted to keep with journal style.
2.6: The authors are making a large number of comparisons based only on 6 replicates. And really they are doing a repeated measures type test if they only have 3 tanks and are sampling each tank twice (to achieve 6 samples). The authors do not provide justification for choosing an ANOVA and do not provide a statement regarding if their data are appropriate for an ANOVA (i.e. meet the assumptions). I do not think a reference for a p value is needed. Also, “p” is generally lowercase. Furthermore, how many females were used for “crude extracts of ovarian tissue”? I ask largely because what they have written is unclear? Did they randomly select two females from each tank? Sample one female from each tank twice? It seems the authors do not do a good job trying to account for if the females they sample even contributed to the embryos they sampled.
3.1: Generally you should not cite a reference in the results section.
3.2. The first few sentences (last paragraph on page 7) are all statements more appropriate to the methods.
L263: What is meant by according to the Ct values? Did you ask them? Probably best to just remove this
as the sentence does not require this clause.
Discussion
L357: I am not certain that I buy Atlantic cod is a pelagic fish.
L367-370: This can be removed. They offer no supporting literature and this is largely unnecessary.
The paragraph about Cathepsin-B could be truncated as some of it is redundant.
L431: Generally do not begin a entence with an abbreviation
Reviewer 2 Report
Very interesting and prospective research. Work written clearly and understandable. In my opinion, it requires only minor supplementary adjustments, which I present below.
Part Introduction. Generally suggestion to supplement this section with information on research topics on freshwater fish as a reference.
Lines from 100 to 104.Suggestion to clearly indicate the aim of the work at the end of this section in a separate paragraph and beginning with words "The aim of this study was..."
Part Materials and Methods.
Lines from 111 to 113. The need to indicate the numbers of specific documents confirming consent to the tests.
Lines from 120 to 121. Necessity to indicate the composition of the feed (mainly protein, fat, dry matter and energy content) and the size of the feed pellets. In addition, supplementing with the amount of the daily food dose. Same data if available for forage fish, squid and cuttlefish.
Lines 172 and 217. A suggestion to change the format of the cited literature to a uniform one throughout the text.
Part Discussion.
Lines from 367 to 370. Suggestion of moving this fragment to the introduction part and combining it with the aim of the work.
Round 2
Reviewer 1 Report
I appreciate the author's careful review of comments and for clarifying the methods.